# Diastereoselective dearomatization of indoles via photocatalytic hydroboration on hydramine-functionalized carbon nitride

Qiao Zhang[1], Wengang Xu [1] ✉, Qiong Liu [2] ✉, Congjian Xia[3], Qi Shao[3], Lishuang Ma[3] & Mingbo Wu [1,3] ✉

A protocol for *trans*-hydroboration of indole derivatives using heterogeneous photocatalysis with NHC-borane has been developed, addressing a persistent challenge in organic synthesis. The protocol, leveraging high crystalline vacancy-engineered polymeric carbon nitride as a catalyst, enables diastereoselective synthesis, expanding substrate scope and complementing existing methods. The approach emphasizes eco-friendliness, cost-effectiveness, and scalability, making it suitable for industrial applications, particularly in renewable energy contexts. The catalyst's superior performance, attributed to its rich carbon-vacancies and well-ordered structure, surpasses more expensive homogeneous alternatives, enhancing viability for large-scale use. This innovation holds promise for synthesizing bioactive compounds and materials relevant to medicinal chemistry and beyond.

Indoles are one of the predominant heteroaromatic skeletons in nature and have wide applications in pigments, fragrances, pharmaceuticals, agrochemicals, and materials science[1–7]. Dearomatizations of planer indole scaffolds have emerged as an important transformation for constructing useful multi-functionalized cyclic products, i.e., indolines with 3d-skeletons[8–12]. Hydroboration of indoles is recognized as an efficient approach, drawing significant attention for its ability to produce boryl indoline derivatives, which act as powerful building blocks in the creation of C–C or C–X bonds through well-developed transformations like the Suzuki reaction[13–18]. This methodology has been extensively applied in the field of medicinal chemistry and material science. Very recently, Ito[19] and Xu[20] independently reported the *cis*-hydroboration of indole via copper-catalyzed hydroboration utilizing $B_2Pin_2$, while this approach is achieved at the expense of consuming strong bases, transition metals, and the diboron reagents (Fig. 1a). Notably, the resulting products from these protocols are limited to *cis*-boryl indoline isomers, primarily due to the instability of the *trans*-isomer and the steric repulsion of the substituents in the intermediates, thereby, the *trans*-isomer is remaining significantly

underexplored. Henceforth, the pursuit of innovative indole hydroboration protocols with better atom economy, complementary diastereoselectivity, and transition metal-free has become highly imperative.

The visible-light-induced C–B bond constructions utilizing *N*-heterocyclic carbene (NHC)-borane via Giese-type hydroboration of unsaturated C–C bonds has become an established strategy from the seminal work of Curran and Wang[21,22], which proceeded with high atomical economy[23–29]. Impressively, Wang and Xie group independently developed the hydroboration of activated alkenes via boryl radical addition to constructing acyclic α or β-boryl carbonyl compounds (Fig. 1b)[30,31]. However, unlike the acyclic alkenes, the addition of a boryl radical to a C=C bond within an aromatic indole system employing NHC-boranes presents a grand challenge that has yet to be investigated. This is primarily due to the strengthened energy barrier resulting from the disruption of the aromatic structures and steric hindrance. Inspired by our independent reports[32], and those from Curran group's that NHC-borane as the boryl radical precursors via hydrogen atom transfer (HAT) could react with electron-deficient

[1]College of New Energy, State Key Laboratory of Heavy Oil Processing, China University of Petroleum (East China), Qingdao, P. R. China. [2]Guangdong Provincial Key Laboratory of Chemical Measurement and Emergency Test Technology (China), Institute of Analysis, Guangdong Academy of Sciences (China National Analytical Center, Guangzhou), Guangzhou, P. R. China. [3]College of Chemistry and Chemical Engineering, China University of Petroleum (East China), Qingdao, P. R. China. ✉e-mail: chmxw@upc.edu.cn; liuqiong@fenxi.com.cn; wumb@upc.edu.cn

**Fig. 1 | Hydroboration and indole dearomatization. a** copper-catalyzed hydroboration of indoles; (**b**) alkene hydroboration with NHC-borane via radical process; (**c**) This work: heterogeneous photocatalytic hydroboration of indole. **PC** photocatalyst; **HAT** hydrogen atom transfer; **Boc** *tert*-butyloxycarbonyl; **NHC** *N*-heterocyclic carbene.

arenes to give dearomatized 1,4-hydroboration products[33], we were motivated to investigate whether similar strategies could be applied to achieve diastereoselective hydroboration of indoles for the construction of previously inaccessible boryl indolines.

On the other hand, there is considerable interest in developing the heterogeneous photocatalyst for facile sustainable use to address the limitations associated with molecular photocatalysts, such as high expenses, unsustainability, non-recyclability, and potential transition-metal contamination. The emergence of metal-free polymeric carbon nitrides (CNs) catalysts has garnered attention due to their merits of visible light response, low-cost and high chemical stability, and the ability to fine-tuned reactivity through electronic band structure and surface modifications[34–37], thus making them attractive appeals. Recently, the utilization of amorphous porosity CNs material has gained attention in photocatalytic C–C or C–heteroatom bond formation reactions[38–42], due to the abundant mesoporous/micropore network, providing increased absorption sites for easier accessibility of the reactants. However, the amorphous framework commonly presents limitations such as short-range structural disorder, resulting in constrained mobility of electrons and holes, and a high density of surface states that promote the annihilation of photoinduced charge carriers[43]. These factors can potentially limit its overall efficiency in various photocatalytic applications. To address these issues, the crafted design of CNs with long-range structural order along with an abundance of surface-active sites become a promising alternative. We envisaged that the use of this finely tuned CNs heterogeneous photocatalyst could substantially pave the way for the development of new atomical and green protocols for achieving the diastereoselective hydroboration of indoles, nevertheless, a concept that has yet to be demonstrated.

Herein, the protocol of transition metal-free heterogeneous-based catalysis to trigger the diastereoselective photocatalytic hydroboration of indoles is reported (Fig. 1c). This method employed a carbon vacancy abundant well-ordered carbon nitride material (**CN-V**) synthesized from a hydramine-functionalized polymerization process. With the cooperative catalysis of **CN-V** and thiols via HAT, the visible-light-induced hydroboration of indole was achieved and produced boryl indolines in surprisingly *trans*-fashion, which complemented copper-catalyzed hydroboration methodologies. It was found the introduction of carbon vacancies and a well-ordered high crystalline structure in **CN-V** not only retarded the recombination of charge carriers but also facilitated the adsorption of indoles and thiols, leading to the easily accessible and simultaneously rapid activation of them. Interestingly, the mechanistic study showed that the intrinsic selectivity for the *trans*-isomer and the base-induced isomerization of the *cis*-isomer cooperatively led to the final selectivity in the cooperative catalytic system of CN-V and HAT.

## Results
### Preparation and characterization of CNs
The polymeric carbon nitride (**CN-V**) photocatalyst with a well-ordered high crystalline structure and abundant carbon vacancies, was prepared involving an ethanolamine (ETA)-assisted pretreatment of dicyandiamide (DCDA) followed by a thermal oxidation step in an air atmosphere. More details were provided in Supplementary Fig. 1. For comparative analysis, another sample of carbon nitride (**CN**) was synthesized using the same procedure applied to DCDA but without the inclusion of ethanolamine. The microstructure of the synthesized **CN-V** was initially examined using scanning electron microscopy (SEM). The SEM image illustrated a network of flake-like structures with stacked layers (Fig. 2a and Supplementary Fig. 2). In comparison to the irregular nanosheets of **CN** (Supplementary Fig. 3), the **CN-V** exhibited a fluffy appearance. The transmission electron microscopy (TEM) image (Fig. 2b) displayed a distinctive sheet-like framework with abundant mesopores. Notably, in contrast to the indistinct diffraction rings observed in the selected area electron diffraction (SAED) of the amorphous crystal structure **CN** (Supplementary Fig. 4), the SAED image of **CN-V** (inset in Fig. 2b) displayed several prominent diffraction rings and obvious crystal lattice fringes in **CN-V** (Fig. 2c)[44]. This suggests a certain level of well-ordered crystal and long-range ordered structure in **CN-V**. In addition, the results from the energy-dispersive X-ray spectroscopy (EDS) elemental mapping images (Fig. 2d) exhibited the distributions of C and N elements throughout **CN-V**. Atomic force microscopy (AFM) measurements demonstrated (Fig. 2e) that the thickness of **CN-V** was measured to be about 4.9 nm, further confirming the presence of flake-like structures. The specific surface area of **CN-V** (Supplementary Fig. 5) was determined to be 56.1 $m^2 g^{-1}$, with a pore volume of 0.26 $m^3 g^{-1}$, which was slightly smaller than that of **CN** (62.7 $m^2 g^{-1}$ and 0.29 $m^3 g^{-1}$).

The structure of **CN-V** was characterized by X-ray diffraction (XRD) and Fourier transform infrared (FT-IR) spectroscopy. Both the XRD pattern and FTIR spectra of **CN-V** (Supplementary Fig. 6) showed a similar structure with **CN**, featuring the characteristic layered heptazine structure[45]. The full width at half maximum (FWHM) of **CN-V**

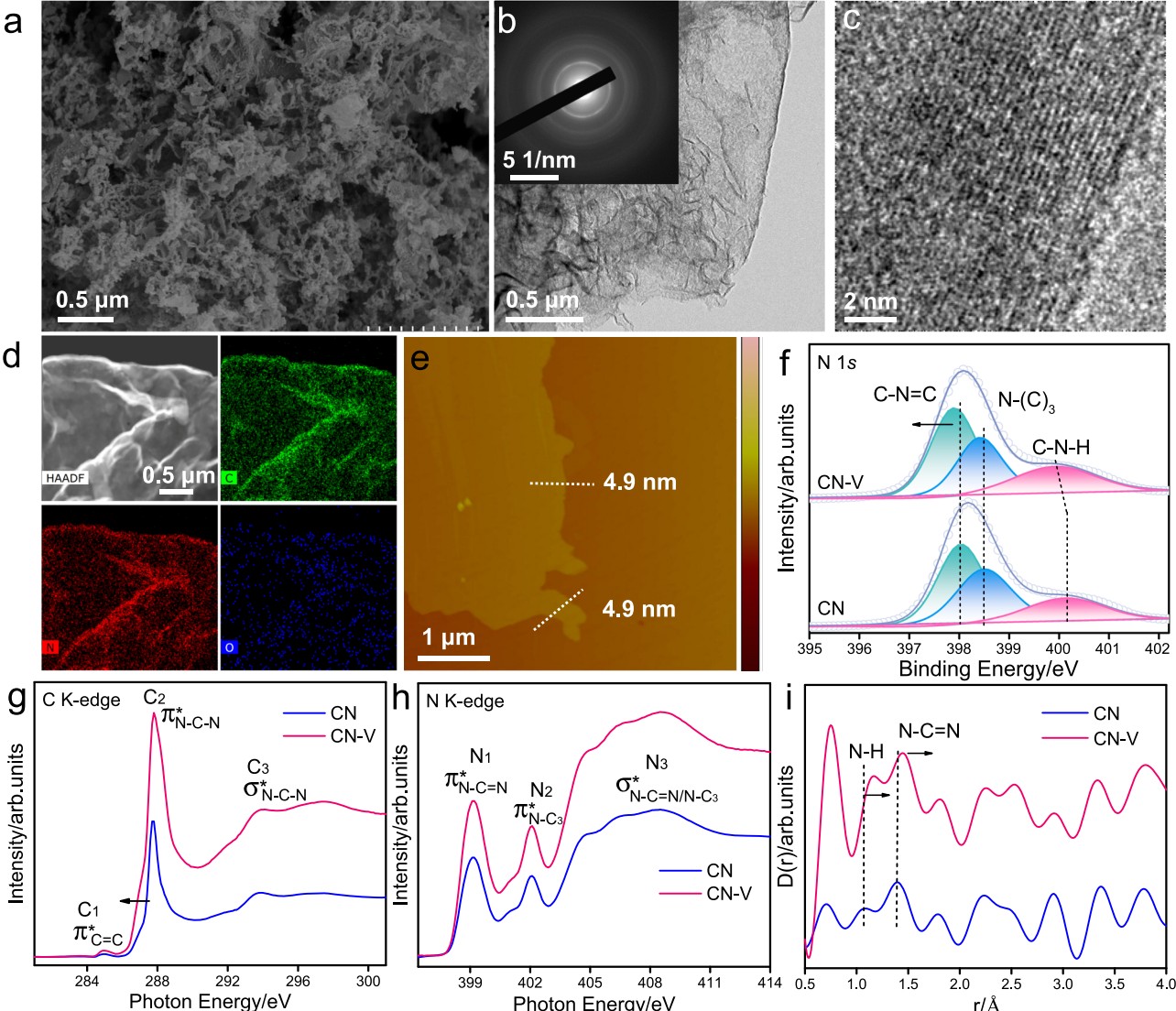

**Fig. 2 | Structural characterization of the photocatalyst. a** SEM image of **CN-V**. **b**, **c** TEM image of **CN-V**, inset in image b is the correspondingly selected area electron diffraction (SAED) pattern. **d** The HAADF-STEM and elemental mapping images of C, N, and O for **CN-V**. **e** AFM image of **CN-V**. **f** High-resolution XPS N 1s results of **CN** and **CN-V**. XANES spectra of **CN** and **CN-V** at the (**g**) C K-edge and (**h**) N K-edge. **i** High-resolution X-ray total scattering spectra with the PDFgui refinement within the structures of **CN** and **CN-V** via the differential correlation function of D(r).

measures 0.577, which is smaller than the FWHM of 0.694 observed in **CN**, the corresponding crystalline diameter based on the Scherrer equation of **CN-V is** larger than **CN**. This suggested a higher crystalline structure of **CN-V**. The chemical compositions and elemental valence states were determined through organic elemental analysis (OEA) and X-ray photoelectron spectroscopy (XPS). The OEA results revealed a measured C/N molar ratio of 0.61 for **CN-V** (Supplementary Table 1), lower than the 0.65 value for **CN**, confirming a relative loss of carbon content in **CN-V**. Moreover, the surface C/N molar ratio, as determined by the XPS results (Supplementary Table 2), decreased from 0.80 for **CN** to 0.75 for **CN-V** consistence with the OEA results. The high-resolution XPS N 1s spectrum of **CN-V** shows three characteristic peaks (Fig. 2f) at 397.9, 398.4, and 399.9 eV, attributed to C−N=C, N−(C)$_3$, and C−NH/NH$_2$. Comparatively, all three N 1s peaks in the **CN-V** spectrum were shifted toward lower binding energies in contrast to **CN**, potentially indicating the generation of carbon vacancies. In addition, the peak area ratios of N-(C)$_3$/C−N=C in the **CN-V**, following ethanolamine (ETA) modification, decreased significantly from 0.82 in **CN** to 0.67, thus implying that a loss of carbon content from the N-(C)$_3$ sites[46]. Moreover, the increase in terminal C−NH/NH$_2$ content

(Supplementary Table 3) signified selective cleavage of the N-(C)$_3$ bond into C−NH/NH$_2$ fragments.

We employed synchrotron-based X-ray absorption near-edge structure spectroscopy (XANES) to further investigate the defect structure of **CN** and **CN-V**. Analysis of the C K-edge XANES spectra (Fig. 2g) revealed two characteristic dipole transition 2p π* resonance signals at 284.9 and 287.8 eV for both samples, corresponding to the defect sites (C$_1$) and N−C−N sp$^2$ hybridization (C$_2$), respectively[44]. The peak at 293.8 eV was ascribed to the transition σ* excitation of the C−N−C bonds. **CN-V** possessed a larger C$_1$ area and higher peak intensity than those of **CN**, signifying a higher carbon vacancy concentration in the **CN-V** structure[44]. In addition, the negative shift (-1.4 eV) in the absorption edge and strengthened white line peak were observed in **CN-V** (Fig. 1g), suggestive of a strong N−C−N interaction and providing evidence of an elevated well-ordered atomic environment structure. Examination of the N K-edge XANES region (Fig. 2h) revealed two transition 2p-π* characteristic resonances are presented at 399.1 eV and 402.1 eV in **CN-V** and **CN**. It can be indexed to the N−C = N (N$_1$) coordination structure in the typical heptazine unit and graphitic-type N-(C)$_3$ (N$_2$) bridging among the neighboring three

**Table 1 | Comparison of the yields for photo-mediated indole hydroboration from various PCs**

| entry | PC | Yield% of 3a[a] | Yield% of 3a'[a] |
|---|---|---|---|
| 1 | **CN-V** | 96[b] | 0 |
| 2 | Ir(ppy)$_3$ (2 mol%) | trace | trace |
| 3 | Ir[dF(CF$_3$)ppy]$_2$(dtbbpy)PF$_6$ (2 mol%) | 72 | 0 |
| 4 | Ir(ppy)$_2$(dtbbpy)PF$_6$ (2 mol%) | 57 | 0 |
| 5 | 4-CzIPN (5 mol%) | 61 | 7 |
| 6[c] | 4-CzIPN (5 mol%) | 0 | 87[b] |
| 7 | **CN** | 63 | 0 |
| 8[d] | **CN-V** | 0 | 0 |
| 9[e] | **CN-V** | 18 | 0 |
| 10[f] | **CN-V** | 20 | 72 |

Reaction conditions: 1-(*tert*-butyl)–3-methyl-1*H*-indole-1,3-dicarboxylate (**1a**, 0.10 mmol), NHC-borane (**2a**, 0.15 mmol), **PC** (5 mg), **ArSH** (30 mol%), K$_2$CO$_3$ (0.05 mmol), DMSO (2 mL), 18 W blue LED irradiation, room temperature, 24 h. [a]Yields were determined by analysis of the crude $^1$H NMR spectra using 1,3,5-trimethoxybenzene as an internal standard. [b]Isolated yields. [c]CH$_3$CN was used instead of DMSO. [d] Without light or **CN-V**. [e] Without **ArSH**. [f] Without K$_2$CO$_3$. 4-CzIPN 2,4,5,6-tetra(9H-carbazol-9-yl)isophthalonitrile.

heptazine units[47]. The enhanced intensity of N$_1$ and N$_2$ in **CN-V** compared to **CN** indicated a relatively enhanced ratio of N species, further indicating an increase in carbon vacancies in **CN-V**. Moreover, the peak ratio of N$_2$/N$_1$ on **CN-V** reduced from 0.49 to 0.46 in comparison with **CN**, further revealing the generation of carbon vacancies located at the N-(C)$_3$ sites, in agreement with the aforementioned XPS results. The atomic environment variation was also explored through high-resolution X-ray total scattering spectra with PDFgui refinement (Fig. 2i). The partial double bond length of C = N−C increased from 1.39 Å (**CN**) to 1.45 Å (**CN-V**), and the N−H bond length is 1.16 Å for **CN-V**, longer than that in **CN** (1.05 Å). These increased bond lengths of both C−N and N−H in the typical heptazine carbon nitride structure unit manifested the generation of vacancies. The enhanced intensity of scattering spectra on **CN-V** manifested the presence of a well-ordered atomic environment structure[44].

**Diastereoselective hydroboration of indoles**

The catalytic activity of this optimized carbon nitride (**CN-V**) serving as the photocatalyst (**PC**) was examined thereafter for the dearomatization of indole derivatives via hydroboration, using 3-substituted indole (**1a**) and NHC-borane (**2a**) as the reaction partners with the cooperation HAT catalyst and bases. A feasible catalytic system with a combination of **CN-V**, **ArSH**, and K$_2$CO$_3$ was developed for the dearomatization of indoles to produce diastereoselective *trans*-hydroboration products (**3a**) in 96% isolated yield, in DMSO under blue LED irradiations (Table 1, entry 1). The noble transition metal-based photocatalysts such as Ir(ppy)$_3$, Ir[dF(CF$_3$)ppy]$_2$(dtbbpy)PF$_6$ and Ir(ppy)$_2$(dtbbpy)PF$_6$ all exhibited lower yields (Table 1, entries 2-4). Interestingly, with 4-CzIPN as the **PC**, the yield and diastereoselectivity were both decreased (Table 1, entry 5). Otherwise, in CH$_3$CN, the *cis*-hydroboration products were selectively obtained (Table 1, entry 6). The structures of **3a** and **3a'** were both confirmed by single crystal XRD analysis (Supplementary Tables 5 and 6). Compared with **CN-V**, **CN** which is prepared without the utilization of ETA provided much lower efficiency for hydroboration (Table 1, entry 7). The presence of light and **PC** was essential for the catalytic system (Table 1, entry 8). Thiols

were also necessary to achieve higher yields (Table 1, entry 9), while the absence of bases led to poor diastereoselectivity for *trans*-isomer (**3a**) (Table 1, entry 10).

**Substrate scope.** After the confirmation of the modified conditions, we further explored the scope of indoles and NHC-boranes (Fig. 3). Indole derivatives bearing varieties of esters at 3-position were good candidates for hydroboration with NHC-borane under standard conditions to give the expected *trans*-boryl indolines (Figs. 2 and 3b–3f) in good to excellent yields. The functional groups such as alkenyl, and terminal alkynyl groups were both tolerated and provided the corresponding borylation products (**3 g** and **3 h**) in excellent yields, without the hydroboration of these unsaturated bonds. Impressively, 3-ester substituted indoles with substituents such as alkoxyl, fluorine, and chlorine at the 5-position were all hydroborylated successfully in moderate to good yields (**3i-l**). Hydroboration of 5-bromoindole afforded the desired products (**3 m**) in only 33% yields because of the production of further hydrodebromination products (**3a**, 33%). Hydroboration occurred smoothly for 6-fluoroindole to give the indoline products (**3n**) in 63% yields. The protecting groups on the nitrogen of indoles could be changed from Boc to Cbz or Bz affording the products (**3o** and **3p**) in slightly lower yields. Interestingly, the substituent at the C3-position could also be acetyl groups to give the 3-acetyl indoline borane derivatives in 40% yields, without the violation of the easily reducible acetyl group (**3q**). With additional substituents such as methyl, fluorine, and chlorine at 5- or 6-positions, the hydroboration still proceeded well to deliver the desired *trans*-indoline products (**3r-t**) in 52%-83% yields. The 3-cyanoindoles could also be dearomatized with NHC-borane to give the *trans*-borylated indoline (**3 u**) in 42% yields. Notably, the hydroboration of indoles bearing 3-amide led to the *cis*-isomer (**3 v**) in 60% yields. Gratefully, natural product-derived indole substrates, such as furfuryl alcohol, geraniol, perillyl alcohol, menthol, and fenchyl alcohol could all be hydroborylated preserving the complex functionalities or stereo-characters to give the indoline derivatives (**3w-aa**) in 37%-82% yields. The B−H bonds in NHC-boranes bearing various N-alkyl substituents including

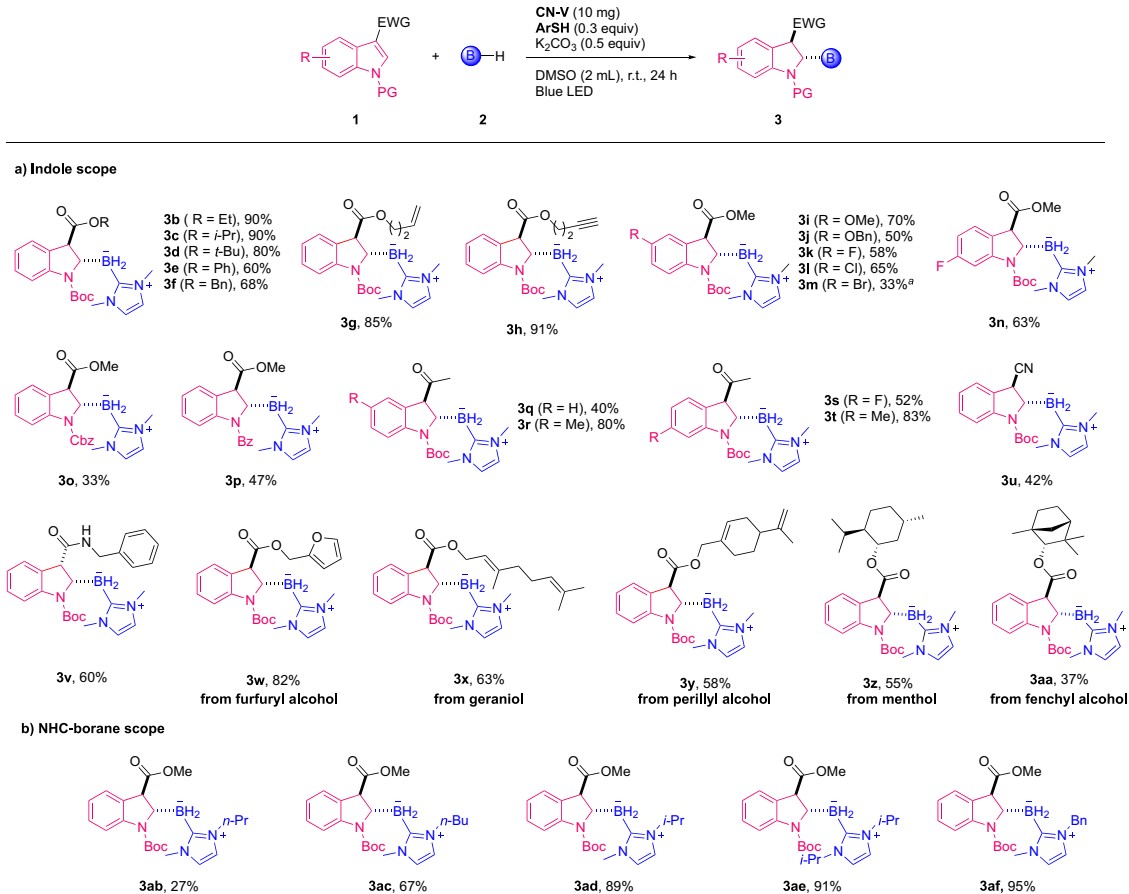

**Fig. 3 | Photocatalytic diastereoselective hydroboration of indoles. a** The substrate scope for indoles. **b** The substrate scope for NHC-boranes. Reaction conditions: indoles (**1**, 0.10 mmol), NHC-borane (**2**, 0.15 mmol), **CN-V** (5 mg), **ArSH** (30 mol%), K$_2$CO$_3$ (0.05 mmol), DMSO (2 mL), 18 W blue LED irradiation, room temperature, 24 h. All cited yields are isolated yields; The product was produced as a single diastereoisomer unless otherwise noted. $^a$The product (**3 m**) was isolated accompanied by the hydrodebromination product (**3a**) with a ratio of 1:1. Cbz benzyl carbonates; Bz benzoyl.

*n*-propyl, *n*-butyl, *i*-propyl, and benzyl were activated and added to indoles to give the indolinyl boranes (**3ab-af**) in 27%–95% yields.

**Synthetic potential.** To further verify the feasibility of this protocol, we conducted the gram-scale experiment with **1a** (4 mmol) under standard conditions to successfully produce indolinyl borane (**3a**) in 83% yields (Fig. 4a). The ester group in **3a** could be further hydrogenated to alcohol products (**4**) in 70% yields[48]. Acylation of this alcohol proceeded smoothly to give the ester (**5**)[49]. Interestingly, the borane in compound **4** could be efficiently transformed into alcohols and delivered the α-hydroxyindole products (**6**)[50]. Notably, all these transformations did not violate the stereo-properties of the starting materials. As a heterogenous catalyst, **CN-V** could be easily recycled with simple filtration. After four times recycling, **CN-V** exhibited consistently high activity. In addition, no obvious structural changes were observed with XRD, FTIR, and SEM characterizations (Supplementary Fig. 7) of the fresh and used catalysts. The same protocol with CN-V could also be applied to the boration of alkenes (**7**) and polyfluoroarenes (**9**) to achieve the products (**8** and **10**) in good yields (Fig. 4b), respectively, exhibiting the great potential of this strategy for the practical organic borane synthesis using NHC-boranes.

## Discussion

To understand the underlying factors of **CN-V** for enhanced photoredox performance, we first conducted the UV-Vis diffuse reflectance spectroscopy (DRS) and Mott-Schottky (M-S) plots to uncover optical absorption properties and the bandgap structure. **CN-V** showed a prominently red-shift absorption edge arising from the π-π* electron transitions and a substantially extended absorption band tail compared to **CN** (Fig. 5a), due to the existence of carbon vacancies and the formation of a higher-ordered crystalline structure. This resulted (Supplementary Fig. 8) in a narrower bandgap of **CN-V** (2.41 eV) compared to that of **CN** (2.83 eV). The M-S plots (Supplementary Fig. 9) revealed that the relatively flat band potential of **CN-V** is −0.82 V[51], which was positively shifted relative to **CN** (−1.00 V). In this sense, the bandgap alignment of **CN** and **CN-V** was proposed in Supplementary Fig. 10. We also tested the redox properties of the indole substrate and **ArSH** using CV (see SI section 7.4). Compared with the position of the conduct and valence band of **CN-V**, the reductive potentials of the indole substrates and oxidative potentials of the thiols were just located between the bandgap, which facilitated the electron transfer process between the photo-excited **CN-V** and the substrates.

DFT calculation for the band structure and density of states (DOS) were employed to understand the role of carbon vacancies. The calculated models shown in Supplementary Fig. 11 and Fig. 5b, c depicted that the bandgap of **CN-V** was smaller than that of **CN** with the conduction band of **CN-V** undergoing a downward shift, which agreed well with the discussed above experimental results. The DOS distributions (Supplementary Figs. 12, 13) indicated that both C 2$p$ and N 2$p$ orbitals contributed to the CB of **CN-V**, while the VB of **CN-V** was mainly composed of N 2$p$ orbitals. The CB position also shifted downwards after the generation of carbon vacancies. With all of this taken into consideration, it was revealed that the introduction of carbon vacancies played an important role in the change in the bandgap of **CN-V**.

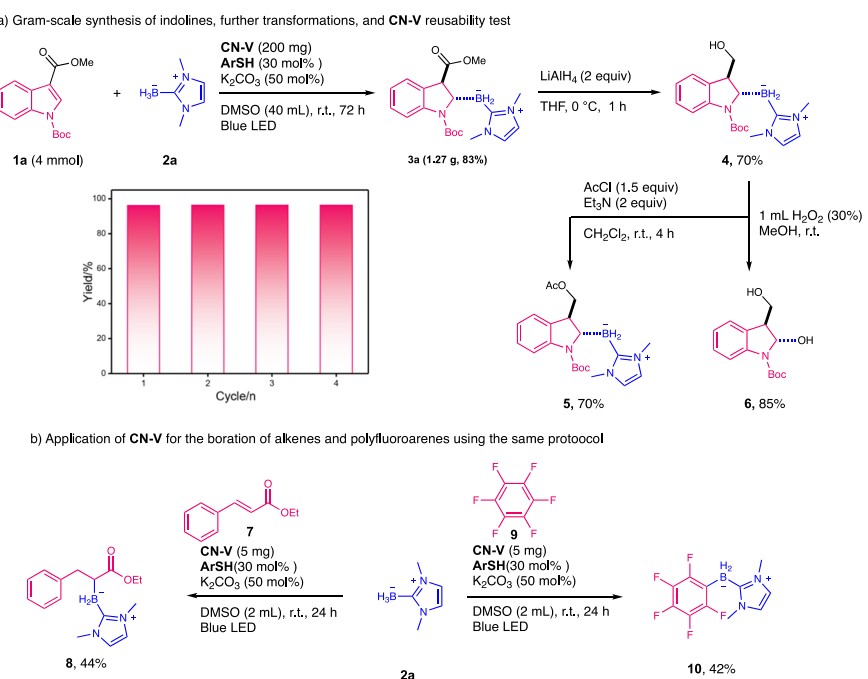

**Fig. 4 | Feasibility tests of CN-V catalyzed organic transformations. a** Gram-scale synthesis of indolines, further transformations, and **CN-V** reusability test. **b** Applications of CN-V for the boration of alkenes and polyfluoroarenes using the same protocol.

To be effectively applied in the photoredox process, the excited carriers should transfer effectively to the surface reactive sites to activate the reactant. The trapping, migration, and transport behaviors of photogenerated carriers in **CN-V** were first investigated by using the time-resolved photoluminescence (PL) spectra. **CN-V** had a shorter average lifetime of 4.0 ns than that of **CN** i.e., 7.6 ns (Fig. 5d), exhibiting rapid charge separation behaviors due to the effective trapping of excited electrons by the carbon vacancies[52]. The solid-state electron paramagnetic resonance (EPR) characterizations (Fig. 5e) presented the EPR intensity of **CN-V** was much more intense compared to that of **CN** in the dark, confirming the raising of delocalization of the electrons and optimized charge separation for **CN-V**[53]. The enhanced intensity after light illumination on **CN-V** manifests the efficient visiting light responsive ability, indicating more excited charge carriers could be generated and efficiently transferred. The enhanced photocurrent density in **CN-V** according to the photocurrent transient response curve (Fig. 5f), further signified the substantially efficient charge separation, rapid transportation due to the introduction of carbon vacancies, and the formation of a higher-ordered crystalline structure.

Moreover, density functional theory (DFT) calculations were employed to further reveal the critical role of carbon vacancies in tailoring the electronic configuration. Fig. 5g, h presented the charge density distribution of the valence band minimum (VBM) and conduction band minimum (CBM) of **CN** and **CN-V**, respectively. The VBM and CBM of **CN** were uniformly delocalized and substantially overlapped along the selected calculated structure cell, which would result in the severe recombination of the photoexcited charge pairs. While, in contrast, the presence of carbon vacancies in **CN-V** led the VBM and CBM to be located in distinct areas. Such a spatially separated charge distribution of VBM and CBM in **CN-V** indicated the retarded recombination of charge carrier pairs. As such, the presence of carbon vacancies favored an efficient charge separation and carrier capture ability, thereby, which was beneficial for increasing photoredox catalytic efficiency. To further probe the effect of carbon vacancies on the absorption and activation of organic reactants, the absorption energy and the electronic charge density difference were calculated. Thiophenol (**ArSH**) and 3-substituted indole (**1a**) were selected as the

model reactant and absorbed on **CN** and **CN-V**. **ArSH** served as a hydrogen atom transfer agent for the vide infra diastereoselective dearomatization of indoles. The adsorption Gibbs energy of **ArSH** is −0.22 eV in **CN-V** (Fig. 5j), lower than −0.06 eV in **CN** (Fig. 5i). This indicated the thiol molecules were easy to interact with **CN-V**, which enabled increased efficiency of charge carrier transportation from **CN-V** to the thiol molecules. We further applied the temperature-programmed desorption (TPD) technology to learn the surface absorption property, as shown in Supplementary Fig. 14. The substantially enhanced peak intensity over the **ArSH**-TPD spectra on **CN-V** and the clearly increased **ArSH** desorption temperature from 119.7 °C on CN to 208.1 °C on **CN-V**, strongly confirmed the better adsorption capability of **ArSH** on the carbon vacancy abundant well-ordered **CN-V**[54]. The 3-substituted indole (**1a**) in **CN-V** presented a −0.20 eV of absorption energy (Fig. 5k), suggesting the facile accessibility of this reactant in **CN-V**. Moreover, the carried charge ($\triangle$q) of **CN-V** after absorbed thiols was calculated to be 0.013 e (Fig. 5l), indicating the efficient charge delivery from **CN-V** to thiols leading to the easy activation of thiols[55]. Collectively, the construction of the carbon vacancy abundant well-ordered carbon nitride material (**CN-V**) would efficiently facilitate the change carrier transportation and enable the facile activation of the indole substrates and oxidation of the thiols with the better utilization of photoinduced electron-hole pairs.

Then, a series of carefully designed experiments were conducted to clarify the reaction mechanism (Fig. 6). By adding TEMPO or vinylcyclopropane to the reaction mixture respectively, luckily, the boryl radical and the N-α-carbon radical were trapped and confirmed with HRMS (Fig. 6a). Using deuterated borane (**2a-D**), the hydroboration product was achieved in 92% yields with only 30% deuteration at the C3-position, indicating the formation of the carbon anion intermediate (Fig. 6b). This conclusion was further verified by the observation of deuteration (60%) at the C3-position after adding D$_2$O to the reaction mixture. In addition, no H-D exchange in the product under the standard conditions occurred with the addition of D$_2$O, implying no further racemization of **3a**. However, when **3a'** as the starting material was used, under standard conditions, 24% yields of **3a** were obtained. In contrast, no racemization was observed without bases. Additionally,

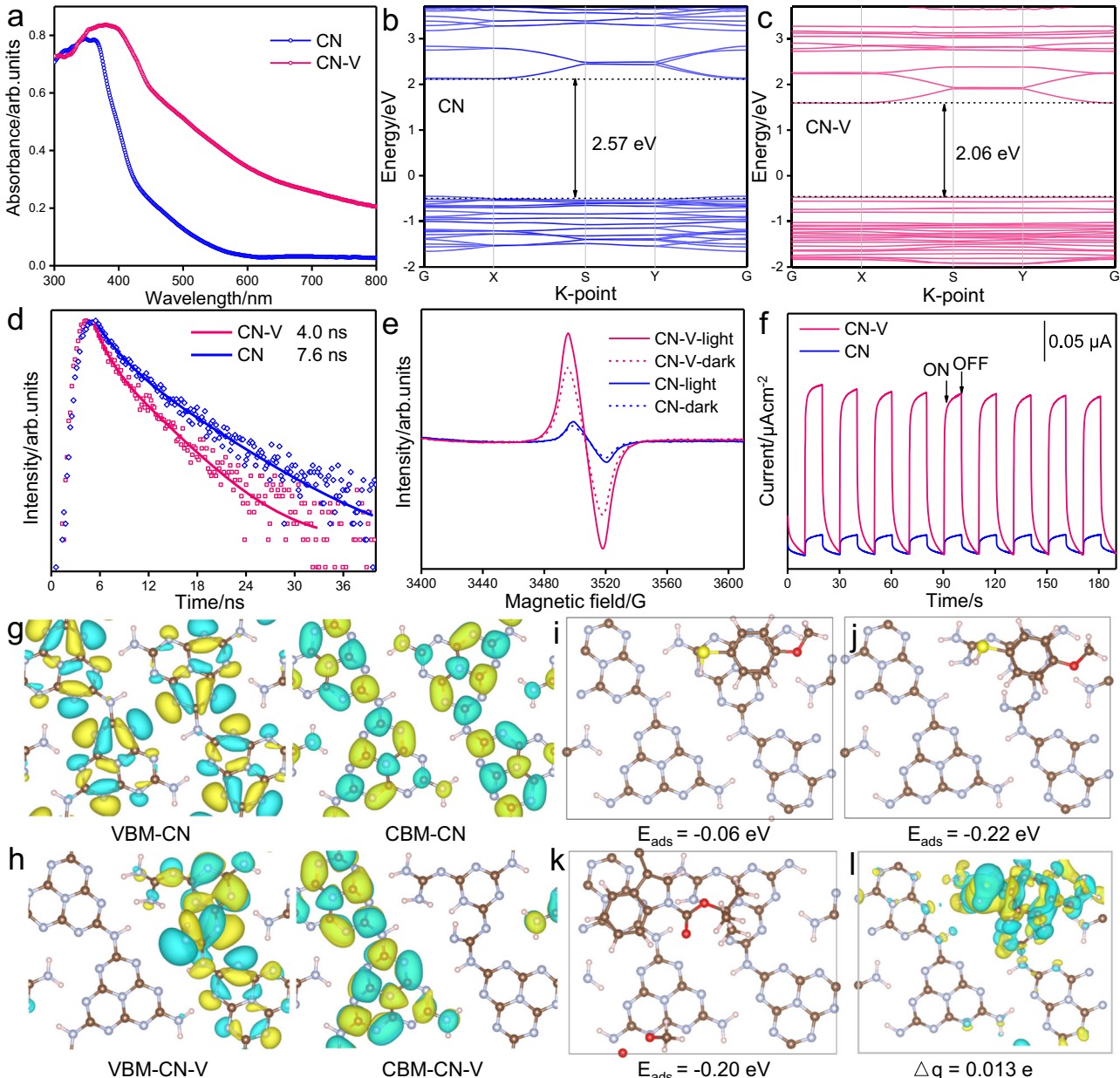

**Fig. 5 | Electronic bandgap structures and charge separation behaviors of CN and CN-V. a** UV-Vis DRS spectra of **CN** and **CN-V**. Calculated band structures of (**b**) **CN** and (**c**) **CN-V**. **d** Time-resolved photoluminescence spectra of **CN** and **CN-V**. **e** Solid-state electron spin resonance (ESR) spectra of **CN** and **CN-V** in the dark time and under illumination for 5 min. **f** Periodic ON/OFF photocurrent response of **CN** and **CN-V** in 0.2 M $Na_2SO_4$ electrolyte under visible light irradiation ($\lambda$ > 420 nm) at an open potential. Charge density based on DFT calculations: (**g**) valence band maximum (VBM) and conduction band minimum (CBM) of **CN**; (**h**) VBM and CBM of **CN-V**. The calculated absorption model and the related energy of ArSH on (**i**) **CN** and (**j**) **CN-V**; (**k**) 3-substituted indole on **CN-V**. **l** Electronic charge density difference of **ArSH** absorbed on **CN-V**.

without **CN-V**, the *trans*-product (**3a**) was obtained in 39% yields from **3a'**, indicating that the photocatalyst may not influence the isomerization process (Fig. 6c). The kinetic study was also performed for this reaction to observe the dual production of the *cis*- and *trans*-isomers in the beginning and the transformation of **3a'** to **3a** after 400 min (Fig. 6d). Above all, the *cis* and *trans*-isomers were both produced during the reaction time and the base was the main promoter for the racemization of **3a'** to achieve the thermodynamic products (**3a**). The Stern-Volmer quenching study for indole and thiols on **CN-V** was also conducted. Thiols (**ArSH**) and indole (**1a**) both showed a quenching effect for **CN-V**. In addition, bases enhanced the quenching effect of **ArSH**. The quantum yields were determined (0.14) to rule out the radical chain mechanism[56].

Based on all the above experimental data, a plausible mechanistic pathway was proposed for the hydroboration of indoles. As illustrated in Fig. 7, after photoexcitation of **CN-V**, thiols (**ArSH**, $E_{1/2}^{ox}$ = + 0.88 V vs NHE in DMSO) or borane ($E_{1/2}^{ox}$ = + 1.16 V vs NHE in $CH_3CN$)[31] itself would be oxidized by the holes at the valence band to thiol radical or boryl radical **II**, respectively. The boryl radical could also be produced via the HAT process of NHC-borane with the thiol radical. Radical addition occurred between the formed boryl **II** and indole substrates (**1a**, $E_{1/2}^{red}$ = − 1.37 V vs NHE in DMSO) to give the radical intermediate (**I**). The electron on the conductive bond was transferred to radical **I** to give the radical anion **II**, followed by protonation, to give the final hydroboration products in both *cis* and *trans*-fashion. Finally, the further transformation of *cis*-isomer to *trans*-products occurred.

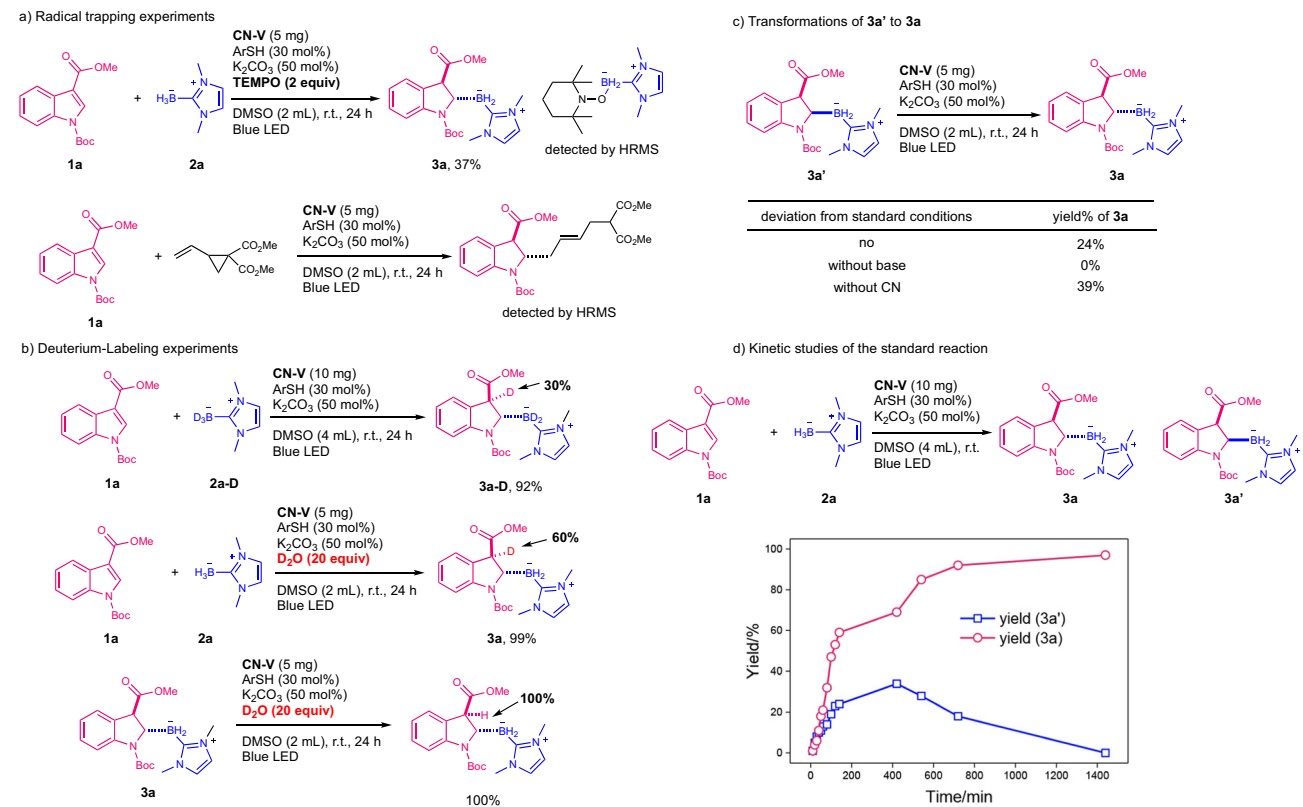

**Fig. 6 | Control experiments and kinetic studies to elucidate reaction mechanism. a** Radical trapping experiments. **b** Deuterium labeling experiments. **c** Transformations of **3a'** to **3a**. **d** Kinetic study of the reaction. TEMPO (2,2,6,6-tetramethyl-1-piperidyloxy).

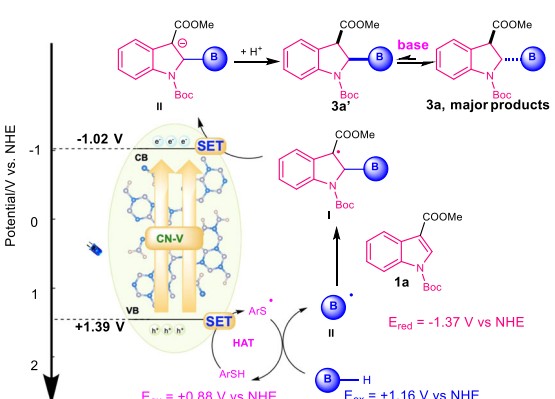

**Fig. 7 | Plausible mechanism.** SET, single electron transfer; HAT, hydrogen atom transfer.

In summary, we have demonstrated a feasible protocol to prepare indolinyl boranes via indole hydroboration with the cooperative heterogeneous photocatalysis and hydrogen atom transfer of NHC-borane. The *trans*-diastereoselectivity was achieved exclusively for hydroboration, which was complementary to the transition metal catalytic protocols. The carbon vacancy abundant well-ordered carbon nitride material exhibited higher efficiency, better reusability, and lower expenses, compared with precious iridium photocatalysts and organic dyes. Varieties of indoles with different functionalities were hydroborated in the transition metal-free and atom-economical manner. The mechanistic study provided evidence that the reaction might proceed in a radical addition pathway and the *trans*-diastereoselectivity was promoted by the bases. Transformations of the indolinyl borane

illustrated the broad applications of this protocol for the preparations of valuable indoline derivatives.

## Methods

### The typical procedure for hydroboration of indole esters

In a 20 mL Schlenk tube with a magnetic stir bar were placed **CN-V** (5 mg), $K_2CO_3$ (6.9 mg, 0.05 mmol, 50 mol%), and NHC-BH$_3$ (**2**, 0.15 mmol, 1.5 equiv). Under nitrogen atmosphere, indole ester (**1**, 0.1 mmol, 1 equiv), 4-methoxybenzenethiol (**ArSH**, 4 μL, 0.03 mmol, 30 mol%), DMSO (2 mL) were added, subsequently. The resulting mixture was sealed and degassed via freeze-pump-thaw three times. Then, the reaction was placed under a blue LED (2-meter strips, 20 W) and irradiated for 24 hrs at room temperature. To the resulting mixture was added water (3 mL), followed by extraction with diethyl ether (5 mL × 3). The combined organic layer was washed with brine (10 mL × 3). The solvent was removed under vacuum. Silica gel chromatography (eluent: Petroleum ether/EtOAc = 2/1) of the crude product afforded the desired compound.

## Data availability

All data generated or analyzed during this study are included in this published paper and its Supplementary Information. Source data are present. All data are available from the corresponding author upon request. Crystallographic data for the structures reported in this Article have been deposited at the Cambridge Crystallographic Data Center, under deposition numbers CCDC 2204094 (3a) and 2204097 (3a'). Copies of the data can be obtained free of charge via https://www.ccdc.cam.ac.uk/structures/. Source data are provided with this paper.

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

## Acknowledgements

We appreciate the help of Dr. Zhang Hongwei from China University of Petroleum (East China) on NMR analysis. We are grateful for the financial support from the National Natural Science Foundation of China (22101301), Fundamental Research Funds for China University of Petroleum (East China) (Grant No. 27RA2014007), China Post-doctoral Science Foundation (Grant No. 31CZ2019010, 05FW2014001), Major Scientific and Technological Innovation Project of Shandong Province (Grant No. 2020CXGC010402) and Shandong Province Natural Science Foundation (Grant No. ZR2020QB043). Dr. Qiong Liu thanks the financial support from the National Natural Science Foundation of China (22309032), Guangdong Basic and Applied Basic Research Foundation (2022A1515011737), Science and Technology Program of Guangzhou (2023A04J1395), and the Guangdong Academy of Sciences Project of Science and Technology Development (2021GDASYL-20210102010)., We thank Dr. Tianxiang Chen and Tsz Woon Benedict Lo (Hong Kong Polytechnic University) for high-resolution X-ray total scattering spectra measurement.

## Author contributions

W.X., Q.L., and M.W. designed this project. W.X., Q.L., Q.Z., C.X., and Q.S. conducted the experiments, analyzed the data, and prepared the Supplementary Information. Q.L. and M.L. performed the DFT calculations. W.X., Q.L., and M.W. analyzed the data and prepared the manuscript.

## Competing interests

The Authors declare no competing interests.
