## [Peer Review File · Nature Communications]

Diastereoselective dearomatization of indoles via photocatalytic hydroboration on hydramine-functionalized carbon nitrideReviewers' Comments:

Reviewer #1:

Remarks to the Author:

This paper reports synthesis of graphitic carbon nitride materials featuring carbon vacancies and application of these materials as heterogeneous photocatalysts in trans-hydroborylation of N-protected indole-3-carboxylic acid esters. The synthesized carbon nitride demonstrates activity comparable to that of certain Ir-polypyridine complex and 4CzIPN. In case of 4CzIPN, diastereoselectivity may be inverted by conducting the reaction in MeCN. Graphitic carbon nitrides were characterized by a series of techniques. The results of DFT modelling revealed that vacancies facilitate sorption of thiol and indole derivative at the carbon nitride basal plane. Overall, the experimental and theoretical results presented in this work are interesting and explain how the C-vacancies in carbon nitride improve activity of this material in organic photoredox catalysis. These results could be expanded to other reactions and other structurally similar materials. However, characterization of the precursor used to prepare CN-V is superficial. It is not clear how and why addition of ethanolamine facilitates introduction of carbon vacancies. The article may be recommended for publication in Nature Communications after a proper revision.

1. Synthesis and characterization of carbon nitride.

1.1. What is the optimal loading of ethanolamine in its blend with DCA in terms of the semiconductor performance in photocatalysis?

1.2. Section 1.1 in the SI, "10 g of dicyandiamide powders dispersed in 100 mL water was heated at 80 °C until it completely melted". Do authors mean that DCDA was dissolved or they actually form deep eutectic mixture?

1.3. According to the experimental procedure, authors prepare and isolate precursor of gCN by heating dicyandiamide with ethanolamine. What is the local chemical/crystal structure of the precursor? Characterization of this substance is required.

1.4. What is the purpose of calcining the precursor twice? How does each calcination step affect the structure of carbon nitride? At what step did author introduce vacancies? What is the activity of carbon nitride prepared in each calcination step in the hydroborylation reaction?

2. Mechanism study:

2.1. What are the redox potentials of the NHC-borane 2a? Being a mesoionic compound, it is expected that sorption of this molecule at carbon nitride is more preferable than that of thiophenol. Therefore, the photocatalytic reaction may be triggered upon electron transfer between this compound and carbon nitride excited state.

2.2. Oxidation potential of thiol (+0.45 V vs SHE) identified by the authors is lower than the reported in literature. More realistic value is $E_{p1/2} = +1.25$ (Figure S17).

2.3. CV measurements of pure electrolyte are suggested to confirm that the peak at $\sim +0.5V$ does not belong to any potential impurities in the DMSO.

2.4. In addition, measurements in MeCN should be conducted to complement those in DMSO. These measurements will also allow excluding the peak at $\sim +0.5 V$ to that belonging to oxidation of a possible DMSO-thiophenol adduct.

2.5. Cathodic current at $\sim -0.85 V$ (Figure S17) might belong to O₂ reduction to superoxide. To determine correctly the reduction potential of the pyrrole derivative, first of all, CV measurements of pure electrolyte saturated with O₂ in the range of potentials +2...-2 V must be conducted. Secondly, electrolyte must be degassed by purging with Ar followed by few cycles in the same range of potentials to ensure that it is free of O₂. After that the substrate must be added and the measurements repeated.

3. In case of photocatalysis with CN-V, CN or Ir-polypyridine complexes, do author also observe inversion of diastereoselectivity when conduct the reaction in DMSO?

4. As indicated by authors, addition of NHC-borane leads to the disruption of the aromatic structure. Is the studied reaction uphill or downhill? I.e. do authors store energy in the products or energy of photons is used to overcome activation barrier? Authors are suggested to calculate Gibbs free energy change in the studied reaction.

5. The elemental mapping images are not atomically-resolved. Therefore, it is not surprising that C

and N elements demonstrate uniform distribution.

6. Page 9. It seems there is a typo in "... on CN-V mi manifest ...". Please correct.

7. In what solvent, did authors conduct their experiments? In the reaction scheme to Table 1 it is written that DMSO was used, while in the footnote – it is THF.

8. Diameter of crystallites may be estimated using the Scherrer equation instead of providing FWHM values.

9. Authors could compare (redox) properties of Ir-polypyridine complexes, 4CzIPN and the synthesized carbon nitrides to support further the proposed mechanism. For example, CN-V, Ir[dF(CF₃)ppy]₂(dtbbpy)PF₆, Ir(ppy)₂(dtbbpy)PF₆ and 4CzIPN mediate the reaction, while Ir(ppy)₃ – does not. Why is it so?

10. It seems that TDOS is greater than the sum of C 2p and N 2p contributions (Figure S10). Why is it so? Are there any other states, which are not shown in the Figure, but contribute to the TSDOS?

Reviewer #2:

Remarks to the Author:

This manuscript by Wu and coworkers describes a heterogeneous photocatalysis to achieve dearomative hydroborylation reaction of indole derivatives with NHC-boranes, providing an efficient protocol to access C2-borylated indolines. Indeed, the paper is certainly detailed and well presented, especially demonstrating a lot of work on the characterization and analysis of CN- to highlight its unique advantages. From a viewpoint of synthetic organic chemistry, hydrogen atom transfer of NHC-borane has been fully established in the radical addition reaction with unsaturated bond (please see some examples: Chem. Commun. 2022, 58, 8380; Adv. Synth. Catal. 2023, 365, 3824-3829; Org. Biomol. Chem. 2022, 20, 3550-3557). In addition, a very similar study has been reported on the dearomative hydroboration of indole derivatives by a combination of photocatalysis and HAT catalysis (please see: DOI: 10.1039/d3qo01452e), which also provides exclusive trans-diastereoselectivity. Compared with previously reported methods, the synthetic value of this paper is not obvious. Therefore, the referee does not consider this work meets the criteria for publication in Nature Communications. However, it is a good piece of work and publication elsewhere would be appropriate with minor changes.

Reviewer #3:

Remarks to the Author:

This manuscript presents a radical hydroboration of indole derivatives via heterogenous photocatalytic strategy. In contrast to previously reported copper-catalyzed hydroboration of indoles where cis-selectivity dominates, this reaction features an exclusive diastereoselectivity for trans-hydroboration product. Using high crystalline vacancy-engineered polymeric carbon nitride (CN-V) as heterogenous photocatalyst, single electron transfer of indole and thiol catalyst took place, ultimately leading to the final product through radical-radical coupling and protonation. Further experiments were conducted to study the photoredox properties of CN-V and probe the underlying reaction mechanism.

Although the manuscript is well organized, the following issues should be addressed in the revision:

- 1) A heterogeneous photocatalytic strategy using cadmium sulfide nanosheets for radical hydroboration reactions has been published by Curran and Dai (Angew. Chem. Int. Ed. 2023, 62, e202306846; not cited) while radical hydroboration of indoles has also been disclosed by An and Wang (Org. Chem. Front., 2024, 11, 149; not cited). Both relevant articles should be cited accordingly.
- 2) It was mentioned in the introduction that the copper-catalyzed hydroborations developed by both Ito and Xu are limited to cis-isomers due to the instability of trans-isomer and steric repulsion of the substituents. Please review both articles to check if the diastereoselectivity arises from diastereoselective protonation or the stability and steric repulsion of the products. If trans-isomer is unstable, what is the plausible reason behind the excellent diastereoselectivity of this reaction?

- 3) The products of this reaction are diastereomeric rather than enantiomeric; hence, bond representations should be revised.
- 4) Referring to a report by Xiang, Chen and Yang (Angew. Chem. Int. Ed. 2020, 59, 6706–6710), hydroboration can occur without thiols as HAT reagents. Have the authors attempted the reaction in entries 2-4 (Table 1) without adding thiols?
- 5) Entries 5 and 6 in Table 1 show a reversal in diastereoselectivity from trans to cis when switching the solvent from DMSO to MeCN with PC = 4-CzIPN. How to explain this result by the proposed mechanism? Does the same phenomenon happen when using CN-V as photocatalyst and MeCN as solvent?
- 6) In entry 10 of Table 1, no cis- product was formed even in the absence of base. This result deserves an explanation in the manuscript to support the proposed mechanism.
- 7) For indole substrates, is the Boc protecting group necessary to increase the product yield? Is it essential to have electron-withdrawing group at C-3 of indole, can common substituents like halogens, amides, or simple alkyl groups be tolerated?
- 8) For the recycled CN-V, have the authors used XRD to show that the catalyst still maintains its highly-order crystalline structure after the reaction?
- 9) Figure 5b: In the third deuterium-labeling (a typo in manuscript) experiment, is the result showing that K₂CO₃ is unable to deprotonate the hydrogen geminal to ester group? In that case how did the product racemize?
- 10) Figure 6: The authors should describe how they obtained both potentials (-1.02 V and +1.39 V) in the manuscript. Besides, how the authors obtain the oxidation and reduction potential of thiol and indole, respectively? The authors should be mentioned in the manuscript whether they are calculated or previously reported to aid the discussion of mechanism.
- 11) The authors suggested that the thermodynamically stable product 3a was obtained under the influence of base. Have the authors used different bases with varying basicity to conduct the reaction? Any mixture of trans or cis isomers was observed?
- 12) Radical-radical cross coupling usually occurs between persistent and transient radicals. How do the authors comment on the possibility of radical-radical cross coupling in the proposed mechanism? Have the authors attempted sterically hindered thiols to see if it affects diastereoselectivity?
- 13) In page 12, "3-nitril indoles" should be "3-cyanoindoles".

Response to Reviewer #1:

Comments:

This paper reports synthesis of graphitic carbon nitride materials featuring carbon vacancies and application of these materials as heterogeneous photocatalysts in *trans*-hydroborylation of N-protected indole-3-carboxylic acid esters. The synthesized carbon nitride demonstrates activity comparable to that of certain Ir-polypyridine complex and 4CzIPN. In case of 4CzIPN, diastereoselectivity may be inverted by conducting the reaction in MeCN. Graphitic carbon nitrides were characterized by a series of techniques. The results of DFT modelling revealed that vacancies facilitate sorption of thiol and indole derivative at the carbon nitride basal plane. Overall, the experimental and theoretical results presented in this work are interesting and explain how the C-vacancies in carbon nitride improve activity of this material in organic photoredox catalysis. These results could be expanded to other reactions and other structurally similar materials. However, characterization of the precursor used to prepare CN-V is superficial. It is not clear how and why addition of ethanolamine facilitates introduction of carbon vacancies. The article may be recommended for publication in Nature Communications after a proper revision.

Response to comment: We are glad to see the reviewer's support for the publication of our work. We are pleased to revise the manuscript according to the comments of the reviewer.

Q1. Synthesis and characterization of carbon nitride.

Q1.1. What is the optimal loading of ethanolamine in its blend with DCA in terms of the semiconductor performance in photocatalysis?

R1.1: According to the Reviewer's suggestion, we prepared a series of CN-V samples solely by varying the amount of ethanolamine added, while keeping the other conditions constant. The specific steps are outlined as follows:

CN-V was synthesized as follows: 10 g of dicyandiamide powders dispersed in 100 mL water was heated at 80 °C until it completely dissolved and then 1, 2.5, 5, and 10 mL of 10 vol.% ethanolamine (ETA) aqueous solution was added. The mixture was kept heating at 80 °C for 2 h, then vaped the water to obtain the faint yellow solid. It was washed with deionized (DI) water, and ethanol and dried at 80 °C under vacuum for 6 h. The solids were then heated to 550 °C /4 h in a tube oven with a heating rate of 7 °C min⁻¹, after naturally cooled to room temperature, further grounded into the fine powder and again calcined at 500 °C for 2 h at a ramping rate of 10 °C min⁻¹ with air atmosphere. The synthesized catalysts by adding 1, 2.5, 5, and 10 mL of 10 vol.% ethanolamine (ETA) are denoted as CN-V₁, CN-V_{2.5}, CN-V₅, and CN-V₁₀.

The catalytic activity of the serial of CN-V serving as the photocatalyst (PC) was then examined for the dearomatization of indole derivatives via hydroboration, using 3-substituted indole (**1a**) and NHC-borane (**2a**) as the reaction partners with the cooperation HAT catalyst and bases, shown in Table R1. Compared to the production of diastereoselective *trans*-hydroboration products with a yield of 63% using CN, the introduction of ETA modification resulted in an improvement of the yields for CN-V. When the concentration of ETA increases gradually from 1 to 2.5 and then 5 mL during the CN-V prepared process, the yields of *trans*-hydroboration products were enhanced, reaching 79% for CN-V₁, 86% for CN-V_{2.5}, and 96% for CN-V₅, respectively. The CN-V₅ exhibited the highest yield among the tested samples. However, excessive ETA addition in CN-V₁₀ led to a decreased yield of 81%. Therefore, when the addition of a 10% volume fraction

ETA aqueous solution is 5 mL, the prepared CN-V₅ exhibits the best performance in photocatalysis process. In this manuscript, unless otherwise specified, CN-V refers to CN-V₅.

entry	deviation	Yield% of 3a ^a	Yield% of 3a' ^a
1	CN-V ₁	79	0
2	CN-V _{2.5}	86	0
3	CN-V ₅	96	0
4	CN-V ₁₀	81	0
5	CN	63	0

Reaction conditions: 1-(*tert*-butyl)-3-methyl-1*H*-indole-1,3-dicarboxylate (**1a**, 0.10 mmol), NHC-borane (**2a**, 0.15 mmol), PC (5 mg), ArSH (30 mol%), K₂CO₃ (0.05 mmol), DMSO (2 mL), 18 W blue LED irradiation, room temperature, 24 h. ^aYields were determined by analysis of the crude ¹H NMR spectra using 1,3,5-trimethoxybenzene as an internal standard.

Q1.2. Section 1.1 in the SI, “10 g of dicyandiamide powders dispersed in 100 mL water was heated at 80 °C until it completely melted”. Do authors mean that DCDA was dissolved or they actually form deep eutectic mixture?

R1.2. Thanks for this question. It means the DCDA was dissolved. We revised the context in the SI.

Q1.3. According to the experimental procedure, authors prepare and isolate precursor of g-CN by heating dicyandiamide with ethanolamine. What is the local chemical/crystal structure of the precursor? Characterization of this substance is required.

R1.3. According to the reviewer’s suggestion, the XRD patterns and FTIR spectra were conducted and compared with those of the DCDA, as shown in Figure R1. The isolate precursor by heating dicyandiamide (DCDA) with ethanolamine (ETA) is denoted as DCDA-ETA. From the XRD patterns, the observed diffraction peaks of DCDA-ETA are largely consistent with those of DCDA, indicating that the crystalline structure of DCDA-ETA is the same as that of DCDA. The FTIR spectra of DCDA-ETA and DCDA are nearly identical, suggesting that the heating of DCDA with ETA does not alter the characteristic structure of DCDA.

Figure R1. (a-b) XRD patterns and (c-d) FTIR spectra of DCDA and DCDA-ETA.

Q1.4. What is the purpose of calcining the precursor twice? How does each calcination step affect the structure of carbon nitride? At what step did author introduce vacancies? What is the activity of carbon nitride prepared in each calcination step in the hydroborylation reaction?

R1.4. Our primary goal was to increase the specific surface area of CN-V, a factor commonly believed to enhance the contact probability between the catalyst and reactants. This was envisioned as a strategy to potentially improve catalytic activity. We then conducted nitrogen absorption-desorption isotherms of CN-V-C1 (only calcinated once) and CN-V (calcinated twice) catalysts to measure the specific surface area (Figure R2). The specific surface area and pore volume of CN-V-C1 is 19.6 m²/g and 0.12 cm³/g, smaller than the CN-V of 56.1 m²/g and 0.26 cm³/g, respectively. We then conducted our investigation with the UV-Vis diffuse reflectance spectroscopy (DRS), the optical absorption band edge of CN-V-C1 and CN-V over 400-470 nm is almost the same. The optical absorption intensity of CN-V among the 480-800 nm is slightly weaker than CN-V-C1. From the FTIR spectra, it is evident that both CN-V-C1 and CN-V present the same main characteristic IR peaks.

Figure R2. (a) The specific surface area, (b) the pore distribution plots, (c) UV-Vis DRS spectra and (d) FTIR spectra of the **CN-V-C1** (only calcinated once) and **CN-V** (calcinated twice).

Note that there have been previous works reporting the etching process using oxidative or reductive gas flow to generate vacancies. However, in this study, both **CN** and **CN-V** catalysts were synthesized by calcinating the precursor twice under identical conditions, with the only difference being that **CN-V** underwent ETA modification. It was verified that **CN-V** has a higher vacancy content than **CN** through XANES and XPS analyses. We are inclined to believe that the addition of ETA plays a crucial role in the formation of vacancies in this work.

In the end, we took the **CN-V-C1** and **CN-V** catalysts to drive the hydroboration reaction under the same conditions. The production of diastereoselective trans-hydroboration products with a yield of 94% using **CN-V-C1**, only slightly lower than the 96% yield by employing **CN-V**. This implies that the twice calcination of the precursor is not the primary factor promoting the enhancement of performance activity; the main factor is attributed to the functionalization of carbon nitride by ETA. But it need to note that the twice calcination of the precursor would enable the slightly advance the performance to some extent.

Q2. Mechanism study:

Q2.1. What are the redox potentials of the NHC-borane **2a**? Being a mesoionic compound, it is expected that sorption of this molecule at carbon nitride is more preferable than that of thiophenol. Therefore, the photocatalytic reaction may be triggered upon electron transfer between this compound and carbon nitride excited state.

R2.1. We thank the reviewer for this valuable suggestion. The oxidative potential of NHC-borane (**2a**) is +0.919 V vs. SCE (refer. 31: Zhu, C. et al. *Angew. Chem. Int. Ed.* **2020**, *59*, 12817.). NHC-borane is the adduct of N-heterocyclic carbene and free borane, which is a covalent compound not showing the properties of a mesoionic compound. In addition, we tested the oxidative potential of thiol to give the oxidative potential of +0.45 V vs SHE. Based on the literature, the deprotonated thiolate salt with an oxidative potential of -0.3 V vs. SCE should be much easier for the oxidation to produce thiol radicals (Shang, R. et al. *ACS Catal.* **2022**, *12*, 4103). In our case, the addition of K_2CO_3 might facilitate the single electron transfer from the carbon nitride excited state to thiols compared with NHC carbene to produce thiol radical and perform further activation of NHC-borane via the hydrogen atom transfer process.

Q2.2. Oxidation potential of thiol (+0.45 V vs SHE) identified by the authors is lower than the reported in literature. More realistic value is $E_{p1/2} = +1.25$ (Figure S17).

R2.2. To confirm the value of oxidative potential of thiol, we rerun the cyclic voltammetry test for thiols and confirmed our results which is +0.88 V vs NHE for 4-methoxybenzenethiol in DMSO (Fig. R3).

Figure R3. Cyclic voltammetry (CV) of **ArSH** in degassed DMSO ($E_{1/2}^{ox} = +0.88$ V vs NHE in DMSO), and pure DMSO electrolyte without **ArSH**.

Q2.3. CV measurements of pure electrolyte are suggested to confirm that the peak at $\sim +0.5$ V does not belong to any potential impurities in the DMSO.

R2.3. We are thankful to the reviewer for this constructive suggestion. We tested the peaks of pure electrolyte (DMSO) and showed that the peak around + 0.5 V disappeared compared with that of the thiol's (**ArSH**) DMSO solutions (Fig. R4), indicating that this peak does not belong to impurities in DMSO.

Figure R4. Cyclic voltammetry (CV) of **ArSH** in degassed DMSO ($E_{1/2}^{ox} = +0.88$ V vs NHE in DMSO), and pure DMSO electrolyte without **ArSH**.

Q2.4. In addition, measurements in MeCN should be conducted to complement those in DMSO. These measurements will also allow excluding the peak at $\sim +0.5$ V to that belonging to oxidation of a possible DMSO-thiophenol adduct.

R2.4. Thanks for this valuable suggestion. We tested the oxidation potential of thiophenol (ArSH) in acetonitrile and the peaks at around $+0.5$ V moved to a more positive position (Fig. R5), illustrating that the previous $+0.5$ V peak might belong to the oxidation of a possible DMSO-thiophenol adduct.

Figure R5. Cyclic voltammetry (CV) of **ArSH** in degassed CH_3CN , and degassed DMSO.

Q2.5. Cathodic current at ~ -0.85 V (Figure S17) might belong to O_2 reduction to superoxide. To determine correctly the reduction potential of the pyrrole derivative, first of all, CV measurements of pure electrolyte saturated with O_2 in the range of potentials $+2 \dots -2$ V must be conducted. Secondly, electrolyte must be degassed by purging with Ar followed by few cycles in the same range of potentials to ensure that it is free of O_2 . After that the substrate must be added and the measurements repeated.

R2.5. Thanks for this important suggestion. We performed the CV test of the pure electrolyte saturated with O_2 in the range of potentials $+2$ to -2 V and found that the reduction potential (-0.85 V) was indeed not that for

substrates, but the oxygen reduction (Fig. R6). Careful CV tests of the degassed solution of indole substrate showed the reductive potential of indole was -1.37 V vs NHE in DMSO, which was higher than the position of the conductive band for excited carbon nitrides (-1.02 V vs NHE in DMSO). In other words, the excited carbon nitrides could not reduce the indole substrates. So, we modified the catalytic cycle in the context (Fig. 6) and the corresponding description of the mechanism from a radical coupling process to a radical addition process.

Figure R6. Cyclic voltammetry (CV) of **1a** in degassed DMSO ($E_{1/2}^{\text{red}} = -1.37$ V vs NHE in DMSO), and O_2 saturated DMSO solution.

Fig. 6 Proposed catalytic cycles.

Q3. In case of photocatalysis with CN-V, CN or Ir-polypyridine complexes, do author also observe inversion of diastereoselectivity when conduct the reaction in DMSO?

R3. We are thankful to the reviewer for this constructive suggestion. In the screening table (Table 1), We used CN, CN-V and Ir-PC as the catalyst for this transformation, we only achieved the *trans*-isomer (**3a**) (Table 1). To confirm the inversion process of the diastereoselectivity, we do the kinetic study using CN-V as the catalyst under standard conditions in DMSO, which exhibited the inversion of the *cis*-isomer (**3a'**) to *trans*-isomer (**3a**). In addition, using the pure **3a'** for the inversion test further indicates the occurrence of this inversion process, which was mainly promoted by the bases (Fig. 5).

Q4. As indicated by authors, addition of NHC-borane leads to the disruption of the aromatic structure. Is the studied reaction uphill or downhill? I.e. do authors store energy in the products or energy of photons is used to overcome activation barrier? Authors are suggested to calculate Gibbs free energy change in the studied reaction.

R4. We thank the reviewer for this suggestion. We conducted the DFT calculation of the Gibbs free energy change for the radical addition process which led to the disruption of the aromatic structure (Fig. R7). The reaction itself is an uphill process which indicates the energy of photons might be used to overcome the activation barrier.

Figure R7. Gibbs free energy change for the radical addition process.

Q5. The elemental mapping images are not atomically-resolved. Therefore, it is not surprising that C and N elements demonstrate uniform distribution.

R5. Thanks for this important comment. Due to the CN-V being a metal-free polymer semiconductor, it is very difficult to successfully gain atomically-resolved elemental mapping images. So, we revised the related content.

Q6. Page 9. It seems there is a typo in "... on CN-V mi manifest ...". Please correct.

R6. We thank the reviewer for this suggestion. We have corrected it to "... on CN-V manifested ..." in the context.

Q7. In what solvent, did authors conduct their experiments? In the reaction scheme to Table 1 it is written that DMSO was used, while in the footnote – it is THF.

R7. We thank the reviewer for this suggestion. We have corrected the footnote based on the experiment we conducted from THF to DMSO.

Q8. Diameter of crystallites may be estimated using the Scherrer equation instead of providing FWHM values.

R8. Thanks for this comment. We then used the Scherrer equation to calculate the diameter of crystallites as follows:

The equation is given by: $D = \frac{K\lambda}{\beta \cos(\theta)}$

Where:

- D is the average crystallite size,
- K is the Scherrer constant (usually taken as 0.943),
- λ is the wavelength of the X-rays or neutrons used, here we used the Cu $K\alpha_1$ radiation, $\lambda = 0.15418$ nm
- β is the full width at half maximum (FWHM) of the diffraction peak. The full width at half maximum (FWHM) of CN-V measures 0.577, which is smaller than the FWHM of 0.694 observed in CN.
- θ is the Bragg angle, the 2θ of the (002) plane in CN-V and CN is 27.82 and 27.74, respectively.

Referring to CN,

$$D = \frac{K\lambda}{\beta \cos(\theta)} = \frac{0.943 \times 0.15418}{(0.694 \div 180) \times 3.14 \times \cos(27.74 \div 2)} = 12.370 \text{ nm}$$

Referring to CN-V,

$$D = \frac{K\lambda}{\beta \cos(\theta)} = \frac{0.943 \times 0.15418}{(0.577 \div 180) \times 3.14 \times \cos(27.82 \div 2)} = 14.881 \text{ nm}$$

As such, based on Scherrer equation, the crystallite diameter of CN-V is 14.881 nm, larger than the CN of 12.370 nm.

Q9. Authors could compare (redox) properties of Ir-polypyridine complexes, 4CzIPN and the synthesized carbon nitrides to support further the proposed mechanism. For example, CN-V, Ir[dF(CF₃)ppy]₂(dtbbpy)PF₆, Ir(ppy)₂(dtbbpy)PF₆ and 4CzIPN mediate the reaction, while Ir(ppy)₃ – does not. Why is it so?

R9. Thanks for this valuable suggestion. Based on our proposed mechanism to produce borane radicals via the hydrogen atom transfer process, we first need the PC to have the potential to oxidize thiols to the thiol radical. However, the excited Ir(ppy)₃ exhibited high reducing ability ($E^{\text{red}} = -1.823$ V vs NHE) and low oxidative potential (0.217 V vs NHE), which is much lower than that of thiophenol (+0.88 V vs SHE). For Ir-polypyridine complexes, 4CzIPN, and the synthesized carbon nitrides, their excited states all have the potential to perform electron transfer with thiols.

Q10. It seems that TDOS is greater than the sum of C 2p and N 2p contributions (Figure S10). Why is it so? Are there any other states, which are not shown in the Figure, but contribute to the TSDOS?

R10. We thank Reviewer #1 for these valuable comments. We accidentally omitted the state density for the C 2s and N 2s state, then leading the TDOS is greater than the sum of C 2p and N 2p contributions; now we recalculated the DOS and shown in Figure R8 and Figure R9.

Figure R8. The density of states of CN and CN-V.

Figure R9. The density of states of (a) C 2s and C 2p and (a) N 2s and N 2p in CN and (c) C 2s and C 2p and (d) N 2s and N 2p in CN-V.

Response to Reviewer #2:

Comments:

This manuscript by Wu and coworkers describes a heterogeneous photocatalysis to achieve dearomative hydroborylation reaction of indole derivatives with NHC-boranes, providing an efficient protocol to access C2-borylated indolines. Indeed, the paper is certainly detailed and well presented, especially demonstrating a lot of work on the characterization and analysis of CN- to highlight its unique advantages. From a viewpoint of synthetic organic chemistry, hydrogen atom transfer of NHC-borane has been fully established in the radical addition reaction with unsaturated bond (please see some examples: Chem. Commun. 2022, 58, 8380; Adv.

Synth. Catal. 2023, 365, 3824-3829; Org. Biomol. Chem. 2022, 20, 3550-3557). In addition, a very similar study has been reported on the dearomative hydroboration of indole derivatives by a combination of photocatalysis and HAT catalysis (please see: DOI: 10.1039/d3qo01452e), which also provides exclusive *trans*-diastereoselectivity. Compared with previously reported methods, the synthetic value of this paper is not obvious. Therefore, the referee does not consider this work meets the criteria for publication in Nature Communications. However, it is a good piece of work and publication elsewhere would be appropriate with minor changes.

Response to comment: Thank the reviewer for the appreciation of our work. Accordingly, we have added more references on hydroboration of unsaturated bonds with NHC-borane (Chem. Commun. 2022, 58, 8380; Adv. Synth. Catal. 2023, 365, 3824-3829; and Org. Biomol. Chem. 2022, 20, 3550-3557) including the work published on indole hydroboration with the iridium-based PC during the revision process (DOI: 10.1039/d3qo01452e). On our hydroboration of indole, compared to the previous study, we could obtain the opposite diastereoselectivities of the boryl indoline products through reaction condition adjustment. Through more detailed kinetic studies and control experiments, we observed the base-induced *cis*- to *trans*-isomerization process to explain the production of the *trans*-isomer. For the substituent on the 3-position, this protocol could be extended to amide groups and surprisingly only *cis*-product was obtained, which indicated the final *trans*-selectivity was closely related to the pKa for the benzylic C-H bond in the indoline products. These comprehensive studies provided mechanistic foundations for further studies on indole dearomatizations.

To further illustrate the importance of our work, we used our newly developed **CN-V** material as the photocatalyst to perform the boration of other substrates including the hydroboration of alkenes and our previously developed defluoroboration of polyfluoroarenes. Moderate yields of the desired organic boranes were obtained (added as Fig. 3b). Our protocol could be potentially used as a practical strategy to produce boryl radicals from NHC-borane with easily prepared, low cost and recyclable heterogeneous catalyst (**CN-V**).

The study on the relationships between the structure of **CN-V** and the catalytic activity further exhibited the two main factors for the improved performance of the carbon nitride catalysts, which are the crystalline structures and the carbon vacancies. This provided the insight for future development of new heterogeneous photocatalysts.

Response to Reviewer #3:

Comments:

This manuscript presents a radical hydroboration of indole derivatives via heterogenous photocatalytic strategy. In contrast to previously reported copper-catalyzed hydroboration of indoles where *cis*-selectivity dominates, this reaction features an exclusive diastereoselectivity for *trans*-hydroboration product. Using high crystalline vacancy-engineered polymeric carbon nitride (CN-V) as heterogenous photocatalyst, single electron transfer of indole and thiol catalyst took place, ultimately leading to the final product through radical-radical coupling and protonation. Further experiments were conducted to study the photoredox properties of CN-V and probe the underlying reaction mechanism.

Although the manuscript is well organized, the following issues should be addressed in the revision:

Response to comment: We appreciate the supportive comments on our work. The following issues were addressed accordingly.

Q1. A heterogeneous photocatalytic strategy using cadmium sulfide nanosheets for radical hydroboration reactions has been published by Curran and Dai (Angew. Chem. Int. Ed. 2023, 62, e202306846; not cited) while radical hydroboration of indoles has also been disclosed by An and Wang (Org. Chem. Front., 2024, 11, 149; not cited). Both relevant articles should be cited accordingly.

R1. Thank the reviewer for this valuable suggestion. We have added these two references accordingly in the ref. list on hydroboration reactions.

Q2. It was mentioned in the introduction that the copper-catalyzed hydroborations developed by both Ito and Xu are limited to *cis*-isomers due to the instability of *trans*-isomer and steric repulsion of the substituents. Please review both articles to check if the diastereoselectivity arises from diastereoselective protonation or the stability and steric repulsion of the products. If *trans*-isomer is unstable, what is the plausible reason behind the excellent diastereoselectivity of this reaction?

R2. We thank the review's comment. In Ito and Xu's reported system, they developed a copper catalyzed hydroboration reaction of indoles. In the plausible catalytic cycle (Fig. R10), the coordination of complex **B** to the C2–C3 bond of indole followed by the subsequent *syn*-addition of the Cu–B bond to the C2–C3 bond would give C-bound enolate (**D**). The protolytic cleavage of the copper–carbon bond of **D** by *t*BuOH would result in *trans*-product. To release large steric congestion between the Bpin group and LCu, **D** would isomerize into O-bound enolate **E**. To avoid the steric repulsion between the Bpin group and bulky *t*BuOH, the protonation of **E** would take place from the opposite side of Bpin to liberate *cis*-product **2c** and **A** for the next catalytic cycle. The diastereoselectivity should arise from diastereoselective protonation caused by the steric repulsion in the intermediates. We revised the manuscript by adding “in the intermediate” after the description on the steric repulsions.

In Xu's report, they mentioned that the *trans*-isomer with Bpin was not stable and could not be isolated. In our case, with NHC-boryl group, the *trans*-isomer was stable and could be isolated.

Figure R10. The proposed catalytic cycles from Ito's paper.

Q3. The products of this reaction are diastereomeric rather than enantiomeric; hence, bond representations should be revised.

R3. Thank the reviewer for this valuable suggestion. We have modified all the related structures in the manuscript and SI.

Q4. Referring to a report by Xiang, Chen and Yang (Angew. Chem. Int. Ed. 2020, 59, 6706–6710), hydroboration can occur without thiols as HAT reagents. Have the authors attempted the reaction in entries 2-4 (Table 1) without adding thiols?

R4. We are thankful to the reviewer for this constructive suggestion. We tested the PCs in entries 2-4 (Table 1) without adding thiols, the hydroboration product was produced, albeit in significantly lower yields (13-56%).

entry	deviations	yield% (3a) ^a	yield% (3a') ^a
1	Ir(ppy) ₃	13	0
2	Ir[dF(CF ₃)ppy] ₂ (dtbbpy)PF ₆	56	0
3	Ir(ppy) ₂ (dtbbpy)PF ₆	14	0

^a The yield was obtained with ¹H-NMR using trimethoxybenzene as the internal standard.

Q5. Entries 5 and 6 in Table 1 show a reversal in diastereoselectivity from *trans* to *cis* when switching the solvent from DMSO to MeCN with PC = 4-CzIPN. How to explain this result by the proposed mechanism? Does the same phenomenon happen when using CN-V as photocatalyst and MeCN as solvent?

R5. We are thankful to the reviewer for this comment. Entries 5 and 6 in Table 1 exhibited the significant solvent effect for the diastereoselectivity of this transformation. The extensive mechanistic studies in our paper indicated that the *cis*- to *trans*- isomerization process occurred and was mainly promoted by the base added. In addition, bases such as K_2CO_3 have a higher solubility in DMSO, which is beneficial for this isomerization process to produce **3a**. Based on these observations, we proposed that the pKa difference of the bases caused by the solubility in different solvents may led to the *trans* to *cis* with PC = 4-CzIPN. When we used the combination of MeCN as solvent and CN-V as the photocatalyst, the diastereoselectivity was also reversed compared with that in DMSO. The major product was *cis*-isomer (**3a'**, 63% yield) with the minor *trans*-isomer in only 37% yields. This observation further illustrated the solvent effect for the diastereoselectivity.

Q6. In entry 10 of Table 1, no *cis*-product was formed even in the absence of base. This result deserves an explanation in the manuscript to support the proposed mechanism.

R6. We thank the reviewer for this careful revision. We re-examined our initial result and repeated the reaction in entry 10 of Table 1, and found that in the absence of a base, we obtained the *cis* product **3a'** in 72% yields and the *trans* product **3a** in 20% yields. We revised the data in the table and the context accordingly as “Thiols were also necessary to achieve higher yields (Table 1, entry 9), while the absence of bases led to poor diastereoselectivity for *trans*-isomer (**3a**) (Table 1, entry 10)”.

Q7. For indole substrates, is the Boc protecting group necessary to increase the product yield? Is it essential to have electron-withdrawing group at C-3 of indole, can common substituents like halogens, amides, or simple alkyl groups be tolerated?

R7. We are thankful to the reviewer for this constructive suggestion. We tested the indole substrates free of the Boc protecting group for hydroboration under modified conditions leading to no desired product. As shown in Figure 2, the other protecting groups such as Cbz, and Bz, instead of Boc both led to much lower yields. These data indicated that the Boc protecting group is necessary to increase the yields. In addition, we performed the reaction of 3-Br and 3-Me indole derivatives. However, no hydroboration products were produced. The halide and alkyl groups were not tolerated for this transformation. For indole bearing an amide group at C3 position, notably only *cis*-hydroboration product was obtained in 60% yield. We have added this product as 3v in Fig. 6 and modified the other content accordingly. These observations indicated that the electron-withdrawing group at C-3 of indole is essential and could influence the stereoselectivity for hydroboration.

Q8. For the recycled CN-V, have the authors used XRD to show that the catalyst still maintains its highly-order crystalline structure after the reaction?

R8. We thank the reviewer for these valuable suggestions. We conducted the XRD pattern of the used CN-V, and made a comparison with the fresh CN-V (Shown in Figure R11). The diffraction peaks of used CN-V are mostly the same with fresh CN-V, demonstrating the highly-order crystalline structure of CN-V is well maintained after the reaction.

Figure R11. XRD patterns of fresh CN-V and used CN-V.

Q9. Figure 5b: In the third deuterium-labeling (a typo in manuscript) experiment, is the result showing that K_2CO_3 is unable to deprotonate the hydrogen geminal to ester group? In that case how did the product racemize?

R9. We are thankful to the reviewer for this significant comment. In Fig. 5b, the *trans*-isomer (**3a**) was used as the starting material under standard conditions with D_2O (20 equiv.) as the additives, we did not observe any isomerization to *cis*-isomer or deuteration of the hydrogen geminal to the ester group. This result indicated that the deprotonation of *trans*-isomer with K_2CO_3 was much more difficult than that for *cis*-isomer (in Fig. 5c), which might be induced by the steric repulsion of the boryl group with the base to perform deprotonation. The product racemization mainly occurred on *cis*-isomer (**3a'**) to produce the *trans*-product (**3a**), but not in the opposite direction.

Q10. Figure 6: The authors should describe how they obtained both potentials (-1.02 V and +1.39 V) in the manuscript. Besides, how the authors obtain the oxidation and reduction potential of thiol and indole,

respectively? The authors should be mentioned in the manuscript whether they are calculated or previously reported to aid the discussion of mechanism.

R10. We are thankful to the reviewer for this constructive suggestion. The oxidation and reduction potential of thiol and indole are obtained by our test via cyclic voltammetry and related content was added in the context and SI. For the oxidative potential of NHC-Borane, we referred to the data from the literature (Zhu et al. *Angew. Chem. Int. Ed.* **2020**, *59* (31), 12817-12821.) as ref. 31.

Q11. The authors suggested that the thermodynamically stable product **3a** was obtained under the influence of base. Have the authors used different bases with varying basicity to conduct the reaction? Any mixture of trans or cis isomers was observed?

R11. We are thankful to the reviewer for this constructive suggestion. As shown in the table, we tried bases ranging from weak (NaOAc) to strong bases (*t*BuOK). Weaker bases led to the cis-isomer (**3a'**), while stronger bases preferred the *trans*-isomer. As the basicity increased, the selectivity for *trans*-isomer was much better.

entry	bases	yield% (3a) ^a	yield% (3a') ^a
1	t BuOK	49	0
2	NaOAc	0	90
3	DABCO	6	18

^a The yield was obtained with ¹H-NMR using trimethoxybenzene as the internal standard.

Q12. Radical-radical cross-coupling usually occurs between persistent and transient radicals. How do the authors comment on the possibility of radical-radical cross coupling in the proposed mechanism? Have the authors attempted sterically hindered thiols to see if it affects diastereoselectivity?

R12. Thanks for this valuable suggestion. After re-running the CV test of all the model substrates and thiols in this protocol, we concluded that the excited carbon nitride could not reduce the indole to radical anion. We have updated the mechanism to a more reasonable radical addition pathway. Meanwhile, we performed the reaction using different sterically hindered thiols instead of **ArSH**, such as 2-methyl-2-propanethiol, *tert*-dodecylthiol, and ethyl 2-mercaptopropionate as HAT catalysts, and found that the yields and diastereoselectivities were similar (48%, 56% and 46%, respectively). This observation indicated that sterically hindrance of thiols could not affect the diastereoselectivity.

**S1****S2****S3**
entry	RSH	yield% (3a) ^a	yield% (3a') ^a
1	S1	48	0
2	S2	56	0
3	S3	46	0

^a The yield was obtained with ¹H-NMR using trimethoxybenzene as the internal standard.

Q13. In page 12, “3-nitril indoles” should be “3-cyanoindoles”.

R13. Thanks for this comment. We revised the manuscript accordingly.

Reviewers' Comments:

Reviewer #1:

Remarks to the Author:

In a revised version of the manuscript, authors addressed referees' comments and performed some additional experiments. The manuscript can be published in Nature Communications.

Reviewer #3:

Remarks to the Author:

The authors have addressed most points this reviewer raised for the initial submission and I believe that the manuscript can be accepted now for publication.

Response to Reviewer #1:

Comments:

In a revised version of the manuscript, authors addressed referees' comments and performed some additional experiments. The manuscript can be published in Nature Communications.

Response to comment: We are glad to see the reviewer's support for the publication of our work. Thanks for the effort on reviewing this paper.

Response to Reviewer #3:

Comments:

The authors have addressed most points this reviewer raised for the initial submission and I believe that the manuscript can be accepted now for publication.

Response to comment: We appreciate the support from the reviewer on the acceptance of our work. Thanks for the effort on reviewing this paper.